# Image Semantic Segmentation of Underwater Garbage with Modified U-Net Architecture Model

**DOI:** 10.3390/s22176546

**Published:** 2022-08-30

**Authors:** Lifu Wei, Shihan Kong, Yuquan Wu, Junzhi Yu

**Affiliations:** 1Department of Advanced Manufacturing and Robotics, College of Engineering, Peking University, Beijing 100871, China; 2Institute of Software, Chinese Academy of Sciences, Beijing 100190, China

**Keywords:** underwater garbage collection, semantic segmentation, deep learning, U-Net

## Abstract

Autonomous underwater garbage grasping and collection pose a great challenge to underwater robots. To assist underwater robots in locating and recognizing underwater garbage objects efficiently, a modified U-Net-based architecture consisting of a deeper contracting path and an expansive path is proposed to accomplish end-to-end image semantic segmentation. In addition, a dataset for underwater garbage semantic segmentation is established. The proposed architecture is further verified in the underwater garbage dataset and the effects of different hyperparameters, loss functions, and optimizers on the performance of refining the predicted segmented mask are examined. It is confirmed that the focal loss function will lead to a boost in solving the target–background unbalance problem. Eventually, the obtained results offer a solid foundation for fast and precise underwater target recognition and operations.

## 1. Introduction

Nowadays, with the growth of global industries, there has been an enormous increase in the production of plastic garbage. Such garbage has created many problems for the conservation of the ecological environment and caused increasingly serious environmental problems, especially regarding water pollution [1,2]. In general, garbage discharged into the water is sparse and dispersed, and it is a challenge to clean with machinery instead of manual cleaning. Cleaning up underwater garbage takes longer time and costs more than cleaning up aboveground garbage. The underwater garbage, unfortunately, will exist for several years to decades, leaving harmful effects on the water quality [3]. It might entangle some species or be accidentally eaten by aquatic animals [4], causing death and affecting the ecological balance. In addition, ship propellers might be winded and stuck by discarded nets, which results in a dangerous voyage. As a consequence, it is necessary to clean up underwater garbage efficiently.

With the development of robotics, artificial intelligence, and autonomous driving in recent years [5,6], it has become possible to apply intelligent robots to accomplish underwater garbage cleaning. To improve the environmental perception ability of the underwater garbage cleaning robot, the primary technical point is to locate and recognize underwater garbage accurately and efficiently. Therein, the image segmentation method is a better methodology compared with the classical deep-learning-based target detection method [7], for it can compute accurate and refined edges of targets [8]. In virtue of the obtained shape and edge information, the underwater garbage cleaning robot can execute more reasonable and precise operations. With respect to the common underwater garbage, such as plastic bags, ropes, fishing nets, and bricks, there are the following characteristics:Similar underwater garbage varies in size and scale;Some underwater garbage has an unfixed shape;The target only has a limited area in an image; hence, the problem of imbalance between the number of targets and background is prominent.

Underwater images are also affected by many limited conditions. When spreading in the water, light is susceptible to absorption and scattering by the underwater medium. Underwater images suffer from visual degradation problems, which adversely affect image recognition tasks. Turbid water bodies and other compositions in the water can also lead to visual degradation of underwater images. The main visual degradation can be classified as image color attenuation and shift, image turbidity, and low image brightness. These are the problems that need to be solved for underwater image segmentation tasks. To effectively detect underwater targets in different situations, a system with high robustness is necessary for this underwater image segmentation task, while a larger dataset can also mitigate the effects of the complex underwater environment.

Image segmentation tasks can be divided into two categories according to the output results: non-semantic segmentation and semantic segmentation. Non-semantic segmentation outputs edge region, or the contour lines of the segmented target, not including the category information of the target. The active contour method is based on a predefined closed contour, and calculates the actual contour of the target through the energy function. Ge et al. proposed a pre-fitted energy-driven active contour model with adaptive edge indicator functions to accelerate the segment speed and reduce the number of iterations [9]. The level set approach can be used to solve the intensity inhomogeneity problem in real-world images. The adaptive data-driven term can optimize the algorithm’s parameters to segment targets at different sizes and features. The additive bias reduces illumination interference, but this method cannot be used for multiple-colors images [10]. Semantic segmentation outputs include segmented contour and the category of each segmented pixel. Semantic segmentation can be used in multi-category segmentation tasks, and most methods involve applying neural networks. The effectiveness of segmentation depends on the network design and the dataset for training the network, which is suitable for multi-category segmentation tasks for specific targets.

In such a situation, the semantic segmentation method is an appropriate way to modify the underwater garbage detection capability so as to assist the robot in extracting more target edge and shape information as well as improve the detection accuracy of targets of various scales. To accomplish the aforementioned technical goal, the fundamental architecture is established based on the U-Net network, which is widely used for biomedical image segmentation tasks [11]. The primary features of the U-Net architecture are symmetrical channels on both sides and skip connection channels that merge different feature maps from each scale of the network. Such a network uses a multilayer convolutional structure at different scales to allow the entire network to preserve features at different scales. More specifically, small targets will be captured by the high-level layers while the large targets will be captured by the low-level layers, hence this network can achieve an impressive result in pixel-level segmentation on multiscale targets [12]. Considering that the underwater garbage targets in this project desiderate a network which has the capacity to the features on both large and small scales, U-Net is an appropriate candidate.

In this paper, an improved U-Net structure is proposed to accomplish underwater garbage image semantic segmentation. First, the unbalanced loss function focal loss and data augmentation strategy are provided to solve the target–background imbalance problem. Meanwhile, the U-Net backbone network is rebuilt with reference to the VGG16 network structure to solve the network capacity problem in the multitarget segmentation task [13]. Note that the primary contributions of this paper are as follows:The network structure of U-Net is improved specifically for underwater garbage targets with a stronger capacity to conduct multiclass segmentation tasks.The underwater garbage semantic segmentation dataset is established to train and evaluate the proposed network, offering a sturdy support platform.To solve the target–background imbalance problem, the special data augmentation strategy and the focal loss function are tightly combined [14]. Experimental results demonstrate an increase in various evaluation indexes via applying this strategy.

The remainder of this article is organized as follows. Section 2 introduces related works in computer vision and U-Net architecture. In Section 3, the underwater garbage dataset and redesign work of the U-Net architecture are accomplished. Next, the experimental results based on the dataset are detailed in Section 4. Finally, Section 5 discusses the experimental results, and the conclusion and the future work are summed up in Section 6.

## 2. Related Works

In this section, the related works of the image segmentation task are discussed in two parts. Computer vision and deep learning applications in underwater images are mentioned in Section 2.1 and Section 2.2, and a specific architecture, U-Net, for image segmentation is discussed in Section 2.3.

### 2.1. Application of Deep Learning in Image Segmentation

The inchoate image segmentation methods are implemented by detecting the grayscale value and grayscale gradient of the image. Image segmentation based on the region thresholding method is a simple segmentation method that directly calculates the appropriate threshold value for segmentation by the gray value of each pixel point, such as the OTSU method [15]. Some edge-detection-based image segmentation methods execute edge detection with pixel grayscale change or gradient, such as the Canny method [16]. Other segmentation methods, such as watershed algorithms, employ morphology to conduct image segmentation [17]. However, none of these can distinguish specific objectives and achieve satisfactory segmentation results in complex situations.

With a series of breakthroughs in deep learning technology, artificial intelligence methods have more widespread applications in image processing works. Classical computer vision techniques can only detect edges or simple template shapes, relying on manual construction of feature engineering, and cannot detect targets in complex situations. By using deep learning techniques, computers are able to extract features from targets autonomously. Especially in recent years, computers have reached higher accuracy than humans in some specific image processing tasks. Meanwhile, the concerned task of computer vision has evolved from image classification, target detection, and even pixel-level semantic segmentation.

For different purposes, researchers in the deep learning field have proposed a variety of different networks, such as deep convolutional neural networks (DCNNs), LeNet [18], AlexNet [19], ResNet [20], and VGG16 for classification; YOLO [21] and SSD [22] for fast target detection; and fully convolutional networks (FCNs) [23], U-Net, and DeepLab [24] for semantic segmentation. The volume of data varies with different tasks. The classification task outputs a classification number through a fully connected layer, object detection needs to output the class, location, and confidence level information of all targets, while the segmentation task predicts the classification for each pixel of the input image and outputs the prediction as a semantically segmented image mask, which has a larger number of parameters than the other tasks. Therefore, the segmentation task demands higher requirements on the computing hardware.

### 2.2. Application of Deep Learning in Underwater Target Recognition

Deep learning technologies have been widely used in the application of underwater target detection, and have enabled AUVs to perform specific tasks. Such robots were designed to clean the garbage on the ground. Bai et al. designed a vision-based robot to accomplish garbage-picking-up work on the lawn [25]. Considering that the underwater environment, such as time-varying water flow and turbid water, leads to increased image uncertainty, the collection of underwater datasets is more difficult. Lakshmi and Santhanam proposed an underwater image recognition method based on convolutional neural network, and compared the accuracy between binary classifier and multiclass classifier [26]. Deep learning can also be used for underwater sonar image detection tasks, identifying mines as well as manmade targets on the seafloor [27]. Using the GAN network can generate datasets that simulate the underwater environment and reduce the impact of small datasets. Transfer learning can also be applied to increase the performance of training [28].

To solve the problem of visual degradation of underwater images, deep-learning-based methods have been applied to underwater image enhancement tasks. Liu et al. proposed a depth residual model for underwater image enhancement recovery based on generative adversarial networks (GANs) and very-deep super-resolution reconstruction model (VDSR). First, underwater images are generated by Cycle-GAN to expand the training dataset, and a deep residual convolutional neural network combined with VDSR is trained via asynchronous training method [29]. The images processed by the model can recover some color and resolution information to achieve image enhancement and reduce the visual degradation of these images on the visual task.

### 2.3. U-Net Deep Learning Network

U-Net is a typical network architecture for biomedical image segmentation based on a full convolutional network. Such a network has not only achieved good results in medical image processing but has also been widely used in other fields of work, such as road segmentation and defect segmentation [30,31]. The most significant feature of U-Net is its symmetric encoder–decoder structure and the skip connection channels between these symmetric convolutional layers on both sides. These skip connection channels enable the entire network to memorize feature maps at different scales and be more accurate in image segmentation tasks. In the contracting path, the U-Net uses two 3 × 3 convolutional layers and one 2 × 2 max-pooling layer to downsample the image. In addition, in the expansive path, the network uses a deconvolution layer and concentrates with the same-scale layer in the contracting path, and then utilizes two 3 × 3 convolutional layers. It is significant to ameliorate the architecture of U-Net to accomplish an accurate result in the specific segmentation tasks.

## 3. Methods

### 3.1. Dataset and Data Augmentation

An underwater garbage datasetis established, which is divided into the training part and the test part. Each of them consists of the image and real label mask. The dataset contains image data of typical underwater garbage taken by monocular and binocular cameras fixed on the underwater robot, which includes four different types of garbage: bricks, plastic bags, nets, and ropes.

All these images are collected in a water reservoir. The images from the binocular robot are stitched and post-processed with an image resolution of 1280 × 480 pixels, while the image from the monocular underwater robot is 640 × 480 pixels, as shown in Figure 1. Meanwhile, images are various when robots work in different environmental situations; thus, a data augmentation strategy is proposed. The data augmentation strategy employs Pillow, a library for image processing. The dataset is expanded by sliding, rotating, flipping, and cutting. The images used to train include 410 images after being augmented from 205 images and the test dataset includes more than 50 images.

### 3.2. U-Net-Based Deep Convolutional Networks

The U-Net deep convolutional network structure is suitable for segmentation tasks, but improvements need to be applied to match challenging underwater garbage targets. The network structure proposed is shown in Figure 2, which is similar to the standard U-Net structure and consists of symmetric encoder and decoder paths on both sides. To achieve a larger capacity of the network, a deeper encoder path similar to the VGG16 structure is adopted for feature extraction in multiscales. Meanwhile, the decoder path is also modified to maintain symmetry with the encoder path. Similar to in the original U-Net, there are still five connection channels between the encoder path and the decoder path, which transport different feature maps in the encoder path.

For an input image of [512 × 512 × 3], in the encode path, convolutional layers of 3 × 3 with a stride size of 2 and 2 × 2 maximum pooling layers are used in the encoder path for successive convolution and downsampling. Five feature maps of different scales are gathered in this process. In the decode path, 2 × 2 upsampling transposed convolution layers and 3 × 3 convolutional layers are applied for upsampling, and then stacked with the five feature maps of the same scale from the downsampling process. Finally, the output data are obtained after a 3 × 3 × *n* convolution layer, where *n* denotes the number of categories. The network architecture is detailed in Table 1.

The activation function of the network for each convolutional layer is a rectified linear unit (ReLU), which only has linear calculations and costs fewer computational resources than logarithmic ones, spending shorter processing time. The cross-entropy loss function is a common choice as a loss function. However, an imbalance problem between background and target exists in normal segmentation tasks. A loss function that gives different weights for various targets is a better choice. Hence, the focal loss function is applied to solve that issue, which is expressed as
(1)$$\left\{\begin{array}{c} FL\left(p_{t}\right)=-{\left(1-p_{t}\right)}^{\gamma }\log\left(p_{t}\right)\\ p_{t}=\left\{\begin{array}{c} p,y=1\\ 1-p,\mathrm{other} \end{array}\right. \end{array}\right.$$
where $p_{t}$ denotes positive target, and γ represents the weights of different categories. By adjusting its hyperparameters, the network can be tuned to better match the actual garbage segmentation task. Note that the Adam optimizer is used in backpropagation.

## 4. Results

Experimental results of the modified U-Net method are described quantitatively and qualitatively in this section, based on the underwater garbage dataset, to confirm the network’s effectiveness for segmentation tasks. Section 4.1 introduces the basic setting of the experiment. Section 4.2 presents the results of the experiments in various conditions.

### 4.1. Settings

The test dataset used for the experiments was collected in a cistern by an underwater collection with a binocular camera and a monocular camera [32], where the depth of the cistern was about 1.5 m.

A total of 350 images are deployed as a training dataset, 39 images as a validation dataset, and 50 images as a test dataset. All images are unified as 512 × 512 RGB images before being sent into the network. To evaluate the network training performance, the confusion matrix is used to calculate precision, recall, F1-score, and intersection over union (IoU).

The model is trained for a total of 30 epochs using a two-stage training mode. In the first 10 epochs of the freeze training stage, the network is trained with a learning rate of $10^{-5}$. In the remaining 20 epochs unfreeze training stage, the learning rate is $10^{-6}$. Pretrained weights are loaded in network initialization steps to improve the training effectiveness because the feature extraction method is similar, especially when using small datasets. The initial weights are too random if training from zero, while using transfer learning can be a wiser method.

### 4.2. Experimental Results

The network takes 1 min 20 s to train each epoch on GPU, and the complete training takes 40 min (NVIDIA RTX3060). The network is almost converged after 20 epochs, and all these rates are stabilized before 30 epochs, indicating that the network training is finished. The focal loss function is applied to reduce the unbalance problem between target and background pixels. In this training stage, the hyperparameters are set as 1.5:1.2:1.5 for plastic bags, ropes (or nets), and bricks. The evolutions of precision, recall, and IoU during the network’s training process and the loss stabilization process of 30 epochs are shown in Figure 3.

As results show in Figure 3, the network achieves more than 87% precision, more than 95% recall, and more than 85% IoU for each category, as detailed in Table 2. From that result, the rope and net are difficult to segment accurately compared with other categories, and receive the lowest rate in precision and IoU. An ambiguous and complex boundary of these ropes and nets may cause that situation.

The quantitative comparative experiments are conducted with respect to the following conditions: (1) training with original U-Net architecture; (2) replace the focal loss with the cross-entropy loss function; (3) training with compressed images and full-size images; (4) training with the SGD optimizer.

To verify the performance of the improved network, the comparative experiments between the original U-Net and the modified one were conducted. Because the original U-Net can only receive the single-channel grayscale images and generate the binary categories outputs, in this test stage the original network adopts the three-channel input and the multi-categories output, remaining asthe original backbone. Note that the loss function is the cross-entropy loss function. The experimental results are shown in Table 3. Compared with the original U-Net, there exists a 10–20% increase in each index on the test dataset via the modified architecture, which indicates that the improved method is significant for the segmentation task.

Table 4 shows results using cross-entropy as the loss function. There are different reductions in precision, recall, and IoU compared with the proposed method. It indicates the effects of focal loss in solving the target–background unbalance problem and making the networks more sensitive to specific targets by adjusting hyperparameters.

Table 5 and Table 6 show results with compressed input images and full-size input images, respectively. Note that compressed input images are [256 × 256] and full-size images are [1280 × 480]. A compressed input increases the network’s efficiency to 11 FPS but causes a decrease in precision, recall, and IoU. Meanwhile, full-size input decreases the network’s speed to 4.3 FPS but is unable to improve the segmentation results in large size.

Table 7 shows results applying stochastic gradient descent (SGD) optimizer, where the learning rate is $10^{-3}$ in the freeze training step and $10^{-4}$ in the unfreeze training step. The result is similar to training with the Adam optimizer, while the SGD optimizer needs a larger learning rate to achieve effective gradient descent.

Some results from the test dataset are shown in Figure 4 to make qualitative analysis, including input images, real masks, and segmentation outputs.

By comparing the output images with real masks, it can be concluded that the segmentation performance of different categories and conditions is satisfactory. Plastic bags have clear boundaries while the boundaries of nets are complex, resulting inaccurate segmentation. With respect to a small target, such as a brick, if far from the camera, it is harder to detect. Additionally, Figure 5 shows the segmentation results of a binocular camera, and Figure 6 shows the results from a monocular camera. The difference in segmentation results between binocular and monocular cameras also needs to be considered further. As shown in Figure 7, image stitching causes segmentation errors in binocular camera inputs. It is concluded that these errors are caused by insufficient resolution. Therefore, applying specific resolution image inputs on different robots ensures that the images are in the resolution region to balance the efficiency and accuracy, which is an appropriate way to solve this problem.

## 5. Discussion

Computer vision has been applied to more complex tasks based on advancements in artificial intelligence and has achieved better performance than humans in particular tasks. One common area of research interest is self-navigation autonomous robots using vision information. This paper proposes a modified U-net architecture to accomplish efficient underwater garbage semantic segmentation. Compared with other semantic segmentation models, such as DeeplabV3+ [33], PSPNet [34], and SegFormer [35], the U-Net-based structure has a simple symmetric encoder and decoder architecture, making it easier to converge when training on small datasets. Meanwhile, this architecture can use our pretrained weights based on transfer learning to accomplish our underwater garbage semantic segmentation task more efficiently.

Various different conditions are tested and the results are conducted in quantitative comparison in order to evaluate and obtain the best performance of the network. We tested the original U-Net network performance in the existing environment, and the results of experimental results in the test dataset concluded that the modified network is more precise in the underwater garbage segmentation task. The initial goal of this paper was to segment underwater targets in artificial water bodies, such as cisterns, open-air pools, and landscape lakes, and the effectiveness of the method was verified through experiments. In the experimental process, we tested the improved method under various conditions to verify the effect of the focal loss function on the accuracy of small targets and the effect of multiscales input images on the network’s outputs, then proposed the image segmentation method for underwater garbage to achieve good segmentation results.

The results demonstrate that the network has acceptable performance in the tasks. An underwater cleaning robot can finish garbage locating and cleaning work by virtue of this method and can collect garbage without a fixed shape successfully.

There are some limitations to this study. First, the lack of datasets is a major limitation for both this study and other studies about underwater vision. Although the method in this paper has good results for the segmentation task of underwater garbage targets in a cistern, it still cannot accurately detect the targets in the complex natural water environment, which is the main limitation of the current period of this study. Fortunately, increasing numbers of underwater datasets have been released in recent years and such studies on data augmentation methods from existing datasets to simulate underwater images are gradually soaring.

There is still room for further research on the general applicability to various garbage and natural work situations. Future work may focus on dataset construction by collecting images in real-world working conditions and making data augmentation from other garbage to underwater garbage datasets. Second, the speed of the segmentation network is less efficient compared with target detection work. The network runs at only 10 FPS on GPU, far from the 30 FPS required for real-time detection. We will test other novel image semantic segmentation methods in underwater targets and reach a higher speed of image segmentation with the help of edge computing.

## 6. Conclusions and Future Work

In this study, we proposed a modified U-Net network structure to match the multiclass target segmentation task of underwater garbage images. The unbalanced loss function focal loss and data augmentation strategy were provided to solve the target–background imbalance problem. First, we used the monocular and binocular cameras of the underwater collection robot to construct the underwater garbage dataset and modified the architecture of the U-Net to receive three channels input images and multi-categories output. The necessity of the improvement was verified in comparison tests with the original U-Net. Second, we investigated the effects of the loss function, optimizer, and input scale on the network performance and determined the final hyperparameters. Experimental results indicate that the built network can achieve the requirements of the underwater garbage segmentation task.

Future work will be concentrated on two parts. First, we will construct and train the model on a larger dataset. Other state-of-the-art semantic segmentation and instance segmentation methods, such as DeeplabV3+, PSPNet, and SegFormer, will be taken into consideration to improve the performance in the underwater garbage segmentation task. Second, we will improve the network architecture and workflow to increase efficiency, then deploy it on underwater robots for practical testing in complex field scenarios.

## Figures and Tables

**Figure 1 sensors-22-06546-f001:**
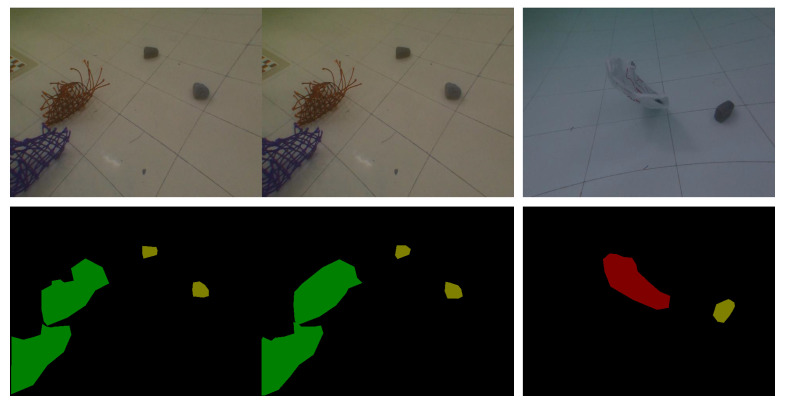
Illustrative examples of the dataset, including two types in each image.

**Figure 2 sensors-22-06546-f002:**
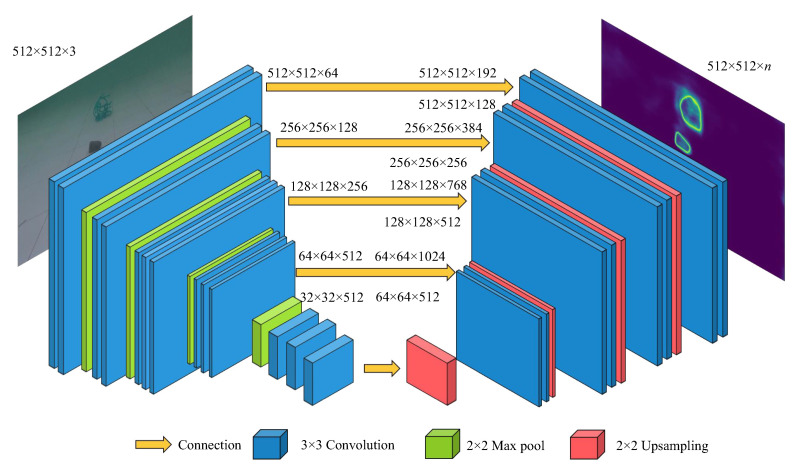
Modified U-Net architecture.

**Figure 3 sensors-22-06546-f003:**
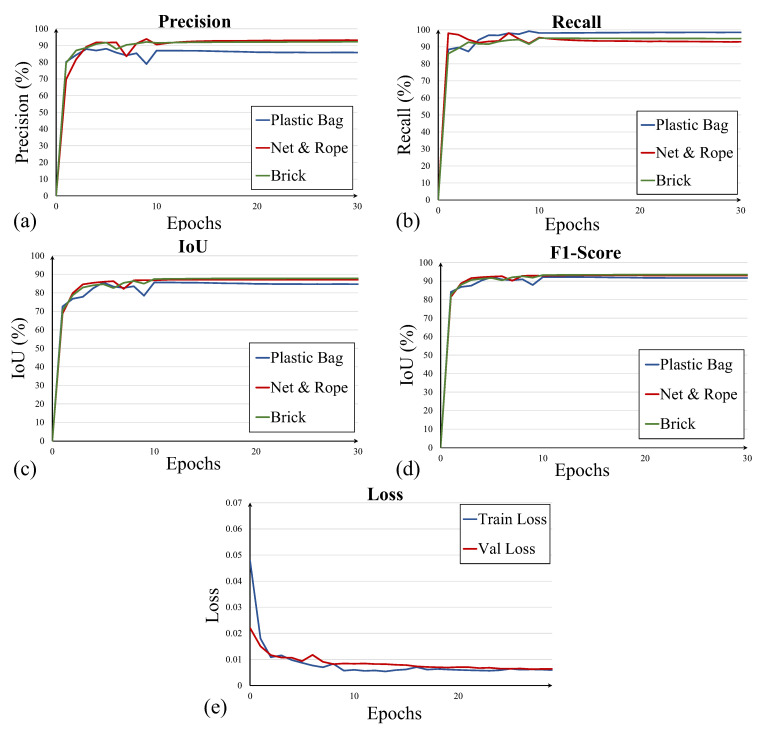
Evolutions of (**a**) precision, (**b**) recall, (**c**) IoU, (**d**) F1-score, and (**e**) the loss stabilization process.

**Figure 4 sensors-22-06546-f004:**
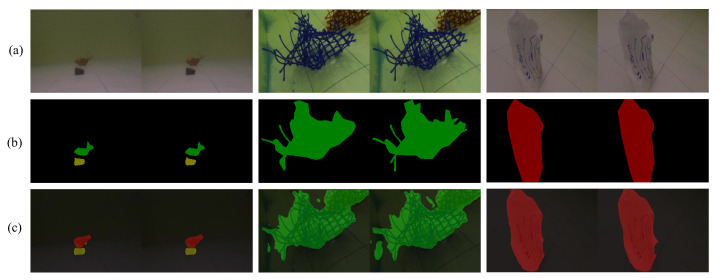
Results of segmentation. (**a**) Input images; (**b**) real masks; (**c**) segmentation results.

**Figure 5 sensors-22-06546-f005:**
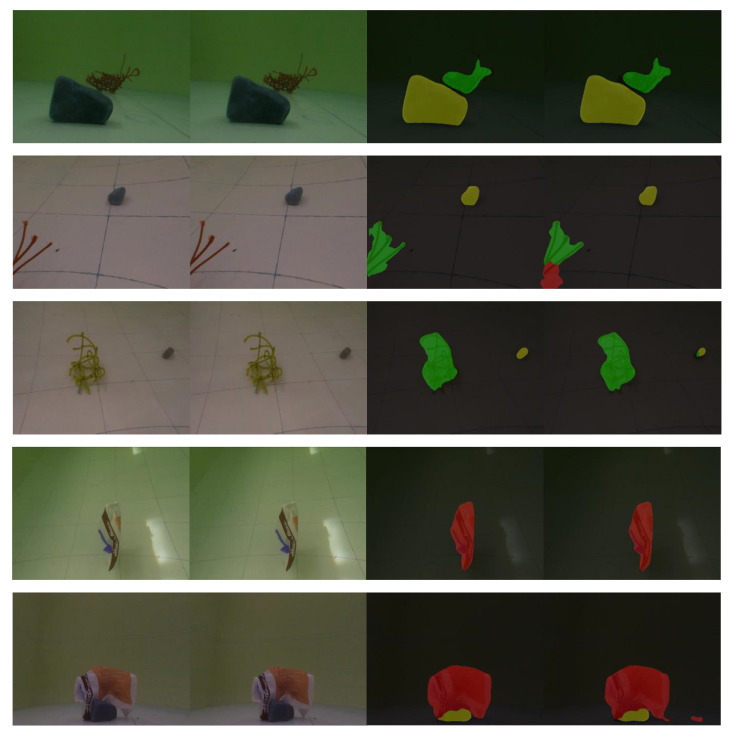
Segmentation results of binocular camera.

**Figure 6 sensors-22-06546-f006:**
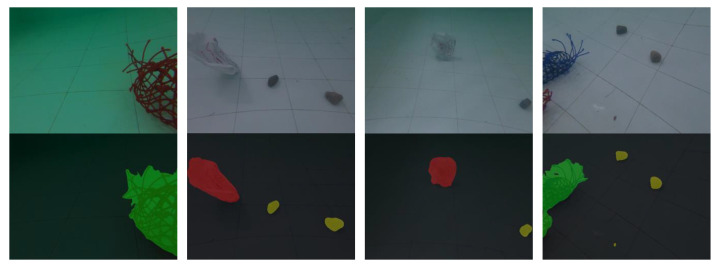
Segmentation results of monocular camera.

**Figure 7 sensors-22-06546-f007:**
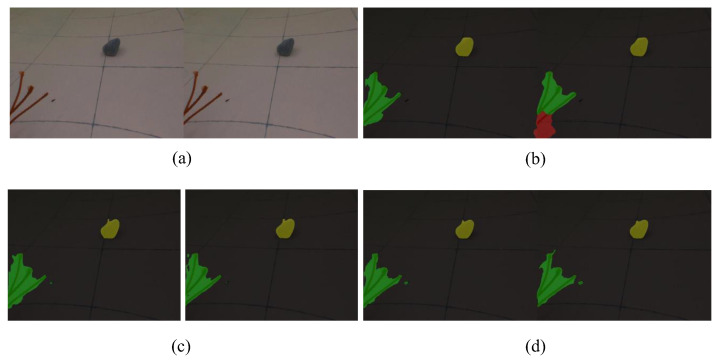
The results of binocular camera input with different resolutions. (**a**) Input image; (**b**) results of stitched binocular camera input; (**c**) results of binocular camera input as two images; (**d**) results of binocular camera input in full-size resolution.

**Table 1 sensors-22-06546-t001:** Architecture of the proposed model.

INPUT		OUTPUT
512 × 512 × 3		512 × 512 × *n*
two 3 × 3 Convolution layers		two 3 × 3 Convolution layers
512 × 512 × 64	Connection	512 × 512 × 192
2 × 2 Max pool layer		2 × 2 Upsampling layer
two 3 × 3 Convolution layers		two 3 × 3 Convolution layers
256 × 256 × 128	Connection	256 × 256 × 384
2 × 2 Max pool layer		2 × 2 Upsampling layer
two 3 × 3 Convolution layers		two 3 × 3 Convolution layers
128 × 128 × 256	Connection	128 × 128 × 768
2 × 2 Max pool layer		2 × 2 Upsampling layer
two 3 × 3 Convolution layers		two 3 × 3 Convolution layers
64 × 64 × 512	Connection	64 × 64 × 1024
2 × 2 Max pool layer		2 × 2 Upsampling layer
two 3 × 3 Convolution layers		two 3 × 3 Convolution layers
32 × 32 × 512	Connection	32 × 32 × 512

**Table 2 sensors-22-06546-t002:** Results of the proposed network.

Method	Proposed Network			
**Time**	**Train: 40 min**		**Predict: 7.5 FPS**	
	**Precision**	**Recall**	**F1-score**	**IoU**
Plastic Bag	0.86	0.98	0.92	0.85
Rope and Net	0.93	0.93	0.93	0.87
Brick	0.91	0.92	0.93	0.88

**Table 3 sensors-22-06546-t003:** Training with original U-Net architecture.

Method	Original U-Net Architecture			
**Time**	**Train: 40 min**		**Predict: 7.5 FPS**	
	**Precision**	**Recall**	**F1-score**	**IoU**
Plastic Bag	0.74	0.74	0.74	0.59
Rope and Net	0.89	0.83	0.86	0.76
Brick	0.70	0.74	0.72	0.74

**Table 4 sensors-22-06546-t004:** Training with cross-entropy loss function.

Method	Cross-entropy Loss Function			
**Time**	**Train: 40 min**		**Predict: 7.5 FPS**	
	**Precision**	**Recall**	**F1-score**	**IoU**
Plastic Bag	0.88	0.86	0.87	0.77
Rope and Net	0.79	0.92	0.85	0.74
Brick	0.92	0.86	0.89	0.80

**Table 5 sensors-22-06546-t005:** Results of compressed input images.

Method	Compressed Input Image			
**Time**	**Train: 15 min**		**Predict: 11.2 FPS**	
	**Precision**	**Recall**	**F1-score**	**IoU**
Plastic Bag	0.76	0.9	0.82	0.69
Rope and Net	0.6	0.87	0.71	0.55
Brick	0.57	0.38	0.46	0.29

**Table 6 sensors-22-06546-t006:** Results of full-size input images.

Method	Full-Size Input Image			
**Time**	**Train: 70 min**		**Predict: 4.3 FPS**	
	**Precision**	**Recall**	**F1-score**	**IoU**
Plastic Bag	0.93	0.96	0.94	0.89
Rope and Net	0.85	0.98	0.91	0.84
Brick	0.92	0.98	0.95	0.90

**Table 7 sensors-22-06546-t007:** Results of SGD optimizer.

Method	SGD Optimizer			
**Time**	**Train: 40 min**		**Predict: 7.5 FPS**	
	**Precision**	**Recall**	**F1-score**	**IoU**
Plastic Bag	0.93	0.95	0.94	0.88
Rope and Net	0.85	0.96	0.90	0.83
Brick	0.94	0.94	0.94	0.89

## Data Availability

Data can be made available upon request from the corresponding author.

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
