# Peer review of "Image Semantic Segmentation of Underwater Garbage with Modified U-Net Architecture Model"

_sensors, 2022, doi:10.3390/s22176546_

Round 1
Reviewer 1 Report
This paper handles Image Semantic Segmentation problem of Underwater Garbage using a Modified U-Net Architecture Model. In general, the topic is interesting, and the comments are listed as follows:
1. The introduction section contain inadequate literature overview and overall content is insufficient.
2. The related work section is also too short and in-comprehensive.
3. The discussion section is too brief and lacks important discussion about experimental results.
4. The conclusion section is too short to conclude this paper and needs more words.
5. The location of some figures are not well-structured while the equations miss commas.
6. Some subsections in this paper are hardly readable due to chaotic and unorganized format.
7. A dataset of 350 images are not enough to train a well-performed neural network.
8. The experiments conducted in this paper sound good, but it needs more details.
9. Other image segmentation methods should also be discussed in the introduction section, e.g., A level set method based on additive bias correction for image segmentation, Expert Systems with Applications; A hybrid active contour model based on pre-fitting energy and adaptive functions for fast image segmentation, Pattern Recognition Letters
Reviewer 2 Report
The work presented in this manuscript is primary on the investigation of different type of camera using the known image segmentation technique for identifaction of objects in the water. The work may be considered for possible inclusion in this journal should the following keypoints be addressed.
(1) The application scenario is for underwater robot to recognize the so-called garbage objects. However, it is not clear to this reviewer if the test conditions are appropriate for real underwater conditions. That is, what is the light intensity, how deep of the objects are allocated under the water, and what is the flow conditions in the real cases? If the test condistions do not match with the real ones, the significance of the proposed method is not justified.
(2) The novelity of the work, in particular the so-called modified U-Net architecture model is not clear. That is, the work presented in the present form is pretty much adjustment of some paramenters and arrangment in the architecutre to recognize objects in the water.
(3) It is not clear from the work what advancement of knowledge or improvement of existing knowledge can be obtained. Frankly speaking, it is more or less an exercise of using the known method in an ad hoc condition from which the authors claimed the method can work for any underwater robots. Also, no comparision of using exisitng methods reported in literature could be clearly found in the manuscript to show improvement of the so-called modified U-Net archetacture than the known ones.
(4) This journal is about the sensors, not the computation related journal. Therefore, the work may not a good fit to the aims of this journal. Consideration of submitting the work to more relavant journal is highly recommended.
(5) Lastly, the work seems to be still in progress. The results reported are not conclusive. Therefore, it is not yet ready at the level of journal quality.
Reviewer 3 Report
This manuscript presents an image semantic segmentation method of underwater garbage with modified U-Net architecture model. The following suggestions can be adopted to improve the manuscript.
1. In section 1, more references should be cited (https://ieeexplore.ieee.org/abstract/document/9056058; https://doi.org/10.1177%2F1729881420976307) to point the study meaning and importance of underwater garbage recognition. Besides, the novelty of this study shoul be further clafied.
2. In section 2, all of the contents are about deep learning, but the application of deep learning in underwater garbage recognition is not mentioned.
3. In section 3, "The images used to train include 410 images after being augmented from 205 images and the test dataset includes more than 20 images", 20 images are too few to test the performance of a deep learning network.
4. I am not sure if the midified U-Net architecture can consistently outperform original U-Net, so the authors can compare the performance of modified and original U-Net networks to varify the necessity of modifing the network.
5. The words in Figure 3 are too small.
6. Some contents have been wildly known by deep learning researchers, such as confusion matrix, equations of precision, recall, f1-score iou, and so on. The reviewer suggest to remove or simplify them.
Round 2
Reviewer 1 Report
The authors have addressed all my comments.
Author Response
We really appreciate your efforts helping us to revise this manuscript, your comments are helpful for us to improve this manuscript and offer valuable inspiration for our further research.
Reviewer 2 Report
My previous questions are properly answered and implemented in the revised manuscript.
Author Response

(The authors gave the same response as above.)

Reviewer 3 Report
Manuscript quality has been improved significantly. However, I has to express several concerns about the revised manuscript:
1. There are many state-of-art semantic segmentation networks (PSPNet, DeepLab V3+, and so on), and the U-Net is not the best one in previous study. Why did the authors choose the U-Net as modified object? The modified U-Net is better than the state-of-art semantic segmentation networks? To verify the performance of the modified U-Net, comparative study of the modified U-Net and the state-of-art semantic segmentation networks is necessary. It is the important highlight in this study. (Note: the hyperparameters of the comparison networks should be optimized.)
2. The reviewer suggests to remove the Table 2 and equation (2)-(5).
3. Why is the f1-score not shown in Figure 3?
